# Nutraceuticals in Pregnancy: A Special Focus on Probiotics

**DOI:** 10.3390/ijms25179688

**Published:** 2024-09-07

**Authors:** Angelica Perna, Noemi Venditti, Francesco Merolla, Sabato Fusco, Germano Guerra, Stefano Zoroddu, Antonio De Luca, Luigi Bagella

**Affiliations:** 1Department of Medicine and Health Sciences “Vincenzo Tiberio”, University of Molise, Via F. De Sanctis, 86100 Campobasso, Italy; 2UO Laboratory Analysis, Responsible Research Hospital, Largo Agostino Gemelli, 1, 86100 Campobasso, Italy; 3Department of Biomedical Sciences, University of Sassari, Viale San Pietro 43/b, 07100 Sassari, Italy; 4Department of Mental and Physical Health and Preventive Medicine, Section of Human Anatomy, University of Campania “Luigi Vanvitelli”, Via Costantinopoli 16, 80138 Naples, Italy; 5Sbarro Institute for Cancer Research and Molecular Medicine, Centre for Biotechnology, College of Science and Technology, Temple University, Philadelphia, PA 19122, USA

**Keywords:** placenta, pre-eclampsia, probiotics, nutraceuticals, placenta development

## Abstract

The placenta is crucial to fetal development and performs vital functions such as nutrient exchange, waste removal and hormone regulation. Abnormal placental development can lead to conditions such as fetal growth restriction, pre-eclampsia and stillbirth, affecting both immediate and long-term fetal health. Placental development is a highly complex process involving interactions between maternal and fetal components, imprinted genes, signaling pathways, mitochondria, fetal sexomes and environmental factors such as diet, supplementation and exercise. Probiotics have been shown to make a significant contribution to prenatal health, placental health and fetal development, with associations with reduced risk of preterm birth and pre-eclampsia, as well as improvements in maternal health through effects on gut microbiota, lipid metabolism, vaginal infections, gestational diabetes, allergic diseases and inflammation. This review summarizes key studies on the influence of dietary supplementation on placental development, with a focus on the role of probiotics in prenatal health and fetal development.

## 1. Introduction

The placenta is a vital organ for the development of the fetus, as it performs essential functions such as nutrient exchange, waste elimination and hormone regulation [1,2,3]. Its development begins at the time of blastocyst implantation and progresses through various stages, which coincide with the chronological stages of embryo development [4,5,6]. It plays a key role in regulating the development of fetal organs, including the heart, brain and kidneys, underlining its importance in ensuring healthy intrauterine fetal growth [7]. Consequently, inadequacies in early development can cause disorders that may not present clinically until the second half of gestation, highlighting the importance of understanding and monitoring early placental development [8]. Defective development of the placenta can lead to disorders such as fetal growth restriction, pre-eclampsia and stillbirth, affecting both short- and long-term fetal health [7,9,10,11]. The mechanisms involved in placental development are highly complex, involving an interaction of maternal and fetal components [12], imprinted genes, signaling pathways, mitochondria and fetal sexomes [13], as well as environmental factors, including nutrition [14], dietary supplementation [15,16] and exercise [17].

In this review we have collected the most relevant studies dealing with the involvement of dietary supplementation on placental development, focusing on the most widely used supplements in recent years and probiotics, which seem to play an important role in prenatal health, placenta development, and fetal development. Our research focused on publications from the last years, using PubMed, Google Scholar, Science Direct and Science Daily as search engines.

## 2. Placental Development

Placental development begins at the time of blastocyst implantation into the uterus and progresses through various chronological developmental stages of the embryo [5]. The placenta is composed of specialized trophoblast cells interacting with extra-embryonic mesenchyme and uterine cells. The cellular processes involved during development are complex and involve various molecular and cellular interactions between uterine and embryonic tissues [10,18,19,20]. Key processes include trophoblast invasion and fusion with endometrium cells, and the establishment of maternal–fetal exchanges [19,21,22,23]. Cell cycle regulatory proteins play a crucial role in the regulation of proliferation, differentiation and invasive capacity of trophoblast cells. The regulation of cell cycle proteins in normal placental development is itself a highly coordinated process involving various mechanisms controlling the timing, localization and activity of cyclins, cyclin-dependent kinases (CDKs) and cyclin-dependent kinase inhibitors (CKIs). In the first phase of development a sufficient number of cells is ensured, in the second phase there is differentiation into cytotrophoblast and syncytiotrophoblast, and finally the invasion of the uterine partition, with the establishment of the primordial maternal–fetal circulation [24]. Signals from the maternal environment, such as oxygen levels and nutrient availability, can also influence the activity of cyclins, CDKs and CKIs, regulating the cell cycle according to the needs of the growing fetus [25]. The failure to develop a normal placenta could be the result of abnormal cell cycle regulation [25]. Transcription factors, which are located in the nucleus and regulate the expression of their target genes within the cell, also play an important role in placental development. They are grouped into a few large families such as zinc finger, leucine zipper, helix–loop–helix, helix–turn–helix and homeobox genes and also include the ligand-activated nuclear receptor superfamily [26]. An increasing number of putative developmental regulators have been identified in the human placenta, but there is little evidence that specific developmental regulators play an essential role in trophoblast differentiation processes such as anchoring villus formation, placental bed invasion or syncytialization. Several homeobox genes with potential roles in controlling commitment and differentiation have been identified in the human invasive trophoblast [26]. Distal-Less Homeobox 3 (*DLX3)*, distal-less homeobox 4 (*DLX4)*, growth arrest-specific homeobox (*GAX)*, human orthologous *Esx1* gene (*ESX1L)* and H.20-Like Homeobox (*HLX)* are key homeobox genes. *HLX* is expressed in proliferating cytotrophoblastic cells during the early stages of placental development, whereas *DLX3* is expressed in the nuclei of villous cytotrophoblastic cells, in the syncytiotrophoblastic layer and in extravillous trophoblastic cells proliferating in the proximal regions of the cell columns both in the first trimester and in the term placenta. *DLX3* regulates expression of human chorionic gonadotropin (hCG) alpha subunit and 3 beta-hydroxysteroid dehydrogenase (3-β-HSD), both of which are important for placental trophoblast function, while several studies have shown that the targets of *HLX* homeobox genes are cell cycle regulatory genes, particularly in trophoblastic and hematopoietic cells. The placenta gradually acquires the ability to perform vital functions such as vascular, respiratory, hepatic, renal, endocrine, gastrointestinal, immune and physical barrier functions, which are critical for the development and safety of the fetus within the maternal environment [5]. Abnormal, flat placentation leads to inadequate utero-placental blood flow, resulting in placental ischemia [27]. This condition reduces the supply of oxygen and nutrients to the fetus, impairing fetal growth and increasing cellular and oxidative stress [28,29,30]. Excessive oxidative stress in the placenta has been linked to the release of factors into the maternal bloodstream that cause widespread endothelial dysfunction and many of the pathophysiological features of pre-eclampsia (PE). Recently, it has been proposed that increased stress in syncytiotrophoblasts may also contribute to PE [31,32,33]. Syncytiotrophoblasts, which form the surface layer of the fetal placental villi, are responsible for transferring nutrients and oxygen to the fetus and for synthesizing and secreting hormones into the maternal bloodstream to support pregnancy adjustments. The exact causes of syncytiotrophoblast stress and placental failure are not fully understood, but are likely to be due to a mixture of genetic and environmental factors, as maternal nutrition playing an important role [34].

Another mechanism that has been shown to be critical for proper placental development is autophagy, an evolutionary response of eukaryotic cells; senescent cells and organelles are targeted to lysosomes for degradation and recycling, maintaining homeostasis of the intracellular environment. The basal level of autophagy in the placental trophoblast has been shown to play an important role during embryo implantation and placental vascular remodeling [35,36,37]. When autophagy is excessive, it changes from a protective to a detrimental mechanism, impairing the metabolic function of placental trophoblasts and leading to certain diseases of placental origin, such as pre-eclampsia and fetal growth retardation (FGR) [38,39,40]. Indicators of this process are microtubule-associated protein l light chain 3 (LC3) and Beclin1, indicators of placental metabolic function [41,42,43]. While some studies have shown that the expression of the autophagy proteins LC3 and Beclin1 is higher in pregnant women with pre-eclampsia than in normal pregnant women [38], others have found no significant differences between pre-eclamptic patients and normal pregnant women by comparing the genes of the placental autophagy pathway [44]. Furthermore, the expression of the autophagic proteins LC3 and Beclin1 was found to be higher in placental tissue from caesarean sections with intrauterine growth restriction than in normal pregnant women [45]. As the pathological process of pre-eclampsia is associated with impaired placental trophoblast cell function and inadequate remodeling of the uterine spiral arteries with oxidative damage, it is possible that placental autophagy activity is increased to help trophoblast cells adapt to pathological changes, maintain bioenergetic homeostasis and eliminate damaged organelles. Thus, women with pathological pregnancies will have increased expression of autophagy-associated proteins in the placenta [46].

## 3. Nutraceuticals

Nutraceuticals are foods or parts of foods that contain bioactive elements with physiological and medicinal effects that provide health benefits and prevent disease. These include prebiotics, probiotics, fibers, fatty acids, antioxidants, spices, herbs, nutrients and supplements. The term “nutraceutical” was coined by Dr Stephen De Felice in 1989, and it is derived from the combination of the words “nutrition” and “pharmaceutical” [47]. Nutraceuticals differ from pharmaceuticals in that they are considered more of a food and less of a drug [48,49,50,51]. The health benefits of nutraceuticals need to be supported by clinical evidence from human studies, particularly the positive effects reported in meta-analyses [15,49,51,52,53,54]. They contain bioactive elements such as polyphenolic compounds, isoprenoids, minerals, amino acid derivatives and fatty acids that have numerous beneficial and healing effects without side effects. They have been shown to have positive effects on the health of the cardiovascular, immune and nervous systems, and play a role in infections, cancer and obesity, making them useful in the prevention of acute and chronic diseases [50,54,55,56,57]. At the moment, regulations on food supplements are not harmonized and vary from country to country. Some countries have no strict regulations, while others have well-structured and strict regulations. Consideration of a harmonized approach to regulation in different countries, based on research, could help increase consumer confidence in dietary supplements worldwide [58]. Currently, regulations in the United States (US), United Kingdom (UK) and Europe are streamlined and conducive to the development of nutraceuticals; India has a nascent regulatory landscape with great potential to compete with other international agencies [59]. However, despite this problem and high product prices, the nutraceuticals market has experienced tremendous growth in recent years, particularly in the US, India and European countries, offering significant opportunities for growth and development [60,61,62]. The most popular nutraceuticals are preparations based on fish oil, prebiotics, probiotics, cranberry, garcinia, ginkgo biloba, ginseng, green tea, omega-3 fatty acids, red yeast rice and turmeric. These products contain key ingredients such as polyphenolic compounds, isoprenoids, minerals, amino acid derivatives and fatty acids [54].

Global figures show that the global nutraceuticals market was worth United States dollar (USD) 498.86 billion in 2021, and is expected to reach USD 1025 billion by 2030, with an estimated annual growth rate of 8.33% [63]. In general, more women take dietary supplements than men, but when looking at athletes, male athletes take significantly more supplements than female ones [64]. Several epidemiological studies also suggest that demographic factors may influence supplement use; older age, gender, education level, affordability and employment status are some prominent demographic characteristics associated with increased supplement use [65]. The Institute of Medicine (IOM) recommends that pregnant women take a prenatal multivitamin/mineral supplement [66]. All women are advised to take a daily supplement containing 30 mg of elemental iron [66] and 600 mg of folic acid daily from a combination of dietary and supplemental sources [66]. However, dietary supplements are not limited to vitamins and minerals. In Canada, for example, vitamin and mineral supplements are classified and regulated as natural health products (NHPs), a classification that also includes herbal remedies, homeopathic medicines, traditional medicines, probiotics, essential fatty acids and amino acids [67]. These products can be purchased without a prescription and are therefore readily available to pregnant women [68] (Table 1).

Probiotics, a term derived from Latin and Greek meaning “for life”, has had several definitions. First defined 50 years ago, the most recent and widely accepted description is “live microorganisms administered in sufficient quantities to exert a beneficial physiological effect on the host”. According to a report by the Joint Food and Agriculture Organization of the United Nations (FAO)/World Health Organization (WHO) Expert Consultation, probiotics are defined as “live microorganisms that, when administered in adequate quantities, have a health benefit to the host”. Many publications define probiotics by referring to similar sources. The International Scientific Association for Probiotics and Prebiotics (ISAPP) also defines probiotics as live microorganisms capable of inducing a health benefit for the host when administered in adequate quantities. The term “probiotics” is often attributed to Lilly and Stillwell, who first coined it in 1965, defining probiotics as a substance produced by one microorganism that stimulates the growth of another. They saw probiotics as the opposite of antibiotics. Parker (1974) offered a different perspective, describing probiotics as organisms and substances that contribute to the intestinal microbial balance. However, the term “probiotics” is often attributed to Kollath, who coined the term “probiotika” in 1953 to describe active substances that are essential for the healthy development of life [69]. Metabolites, dead micro-organisms or other non-viable products based on micro-organisms are not probiotics. There are several types of bacteria used as probiotics, with *Lactobacilli* and *Bifidobacteria* being the most common [70] (Table 2).

These bacteria have a symbiotic relationship with humans and live in the mucus membrane on the epithelial cells of the gut [71]. They inhibit the adhesion and proliferation of harmful bacteria by producing bactericidal chemicals. As evidence of the benefits and safety of probiotics grows, these bacteria are increasingly supplementing and replacing traditional prophylactic and therapeutic regimes. Probiotics are available as supplements containing freeze-dried bacteria in tablets, capsules and powders. Many people use them routinely and the choice of probiotic product depends on the specific type of bacteria and the desired beneficial effect. Thousands of probiotic strains are available, each offering different health benefits.

It has been widely believed that probiotics are not subject to regulation [72]. However, bacteria used as probiotics must undergo thorough safety assessments [73]. The Food and Drug Administration (FDA) has established regulatory authority over the production, manufacturing, labelling and safety of probiotic products. Specifically, on 24 August 2007, the FDA implemented regulations requiring current Good Manufacturing Practices (GMPs) for dietary supplements. Although these regulations do not specifically address the verification of claims of efficacy, they are expected to improve the quality (identity, purity and strength) of probiotic supplements available on the market. In general, the FDA does not review the labelling or safety of probiotic products unless the product is marketed as a drug (i.e., to treat, cure, prevent, mitigate, or diagnose disease) without proper drug approval [72]. FDA regulations on probiotics depend significantly on the intended use, which is indicated on the product label. According to Degnan (2008) [74] and Hoffman et al. (2008) [75], there are four main regulatory categories for probiotics, each with distinct requirements, described in Table 3.

If probiotics are intended to be used as drugs, they are treated as biological products and must comply with FDA regulations applicable to such products. If they are marketed as dietary supplements, they can be sold without FDA pre-approval, but manufacturers must notify FDA of their product claims and comply with new dietary ingredient regulations. If marketed as a food or food ingredient, FDA oversight focuses on post-market controls to prevent adulteration. If marketed as a medical food, pre-market approval is not required under FDA guidelines [74].

In Europe, there are no specific regulations for probiotics used in food, unlike microbial feed additives, which are subject to safety assessments for both animals and humans. The Scientific Committee on Animal Nutrition (SCAN) has introduced the concept of “Qualified Presumption of Safety” (QPS), which allows species with established safety data to be marketed without extensive testing [76].

Probiotics have several health benefits when taken in sufficient quantities: they prevent digestive disorders such as infectious diarrhea, *Helicobacter pylori* infection and antibiotic-associated diarrhea [77,78,79]; they act as an adjunct in the treatment of metabolic disorders such as metabolic syndrome, type 2 diabetes and obesity [78]; they improve gut health, boost the immune response and reduce serum cholesterol levels and blood pressure [80]; they have also been shown to prevent and treat allergies, skin disorders and urogenital infections [81,82]. Probiotics play a key role in modulating the population of the intestinal microbiota, strengthening the intestinal barrier and modulating the immune system [83]; they promote a condition of eubiosis by secreting antimicrobial substances and competing with pathogens to prevent their adhesion to the cell surface [84]. Although probiotics are generally recognized as safe, there are cases where they have been shown to be risky, associated with gastrointestinal disorders, respiratory infections, allergic reactions and serious medical conditions such as sepsis and endocarditis, situations that highlight the importance of assessing the risk associated with taking these products unsupervised [85].

## 4. Nutraceuticals in Pregnancy

Nutrients and nutraceuticals have the ability to influence gene expression and are crucial in preventing diseases by inhibiting detrimental genes [86]. Epigenetics refers to heritable changes in gene expression that do not involve alterations to the DNA sequence itself [87], often resulting from environmental and nutritional factors that modify DNA/histone structures and affect phenotypic expression [86]. There is growing recognition that a mother’s diet during pregnancy impacts the health of her offspring through epigenetic changes [88]. What role might nutraceuticals play in influencing epigenetics during pregnancy? Several studies have looked at supplementation with micronutrients as omega-3 polyunsaturated fatty acids, minerals and vitamins, including folic acid and vitamin D, during pregnancy to see if they might help reduce the risk of conditions such as pre-eclampsia and prevent impaired fetal development, as intrauterine growth restriction (IUGR) [89,90,91,92]. They are the leading cause of perinatal mortality and morbidity worldwide and appear to result from a failure of trophoblast invasion, with critical consequences for placental perfusion by maternal blood [93]. These complications affect both pregnancy outcomes at delivery and the long-term cardiovascular health of affected women and their offspring [94]. Table 4 presents the findings of epidemiological studies conducted by various research groups on the utilization of nutraceutical supplementation during pregnancy in recent years.

In addition to micronutrient supplementation, a large number of pregnant women use many types of extracts for different purposes [95]. The situation varies between Europe and the United States, ranging from 27% to 57% in Europe and 10% to 73% in the United States [96] (in southern Italy, it has been estimated that 81% of pregnant women have used at least one herbal product during pregnancy [97]), while in Australia the percentage is around 14% [98]. The problem with the use of such products during pregnancy is that there are not enough studies to rule out possible effects on the proper development of the placenta and the fetus, so there is no regulation of their use. Many experimental studies have been carried out to analyzing the effects of extracts of various herbs in animal models of mice or rats, but the results have often been contradictory [99,100,101]. Many of these products affect the cytochrome P450 superfamily (CYP), which is responsible for 65-80% of all CYP-mediated drug metabolism [102]; black elderberry, ginger and horsetail show potent inhibition of CYP1A2, while fennel and raspberry leaves inhibit CYP2D6 and CYP3A4 [103].

The use of probiotics by pregnant women is increasing and is generally safe and well tolerated, although some rare side effects have been reported, systemic absorption in healthy women is rare. The current literature does not indicate an increase in adverse pregnancy outcomes [104].

**Table 4 ijms-25-09688-t004:** A review of the literature on the effects of nutraceuticals during pregnancy [105,106,107,108].

Nutraceutical	Key Findings	Reference
Folic Acid	The timely initiation of folic acid (FA) supplementation during gestation was associated with a decreased risk of congenital malformations, which was mainly attributed to its protective effect against heart defects. It is recommended that FA supplementation should be initiated 1.5 months prior to conception and continued for a period of four months in order to optimize the prevention of congenital malformations.	Dong, Jing et al., 2023 [105]
Omega-3 Fatty Acids	It is recommended that pregnant women consume an additional intake of at least 100-200 mg/d of docosahexaenoic acid (DHA), as advised by the European Food Safety Authority (EFSA). Observational studies have demonstrated that a reduction in omega-3 DHA and eicosapentaenoic acid (EPA) intake and a decline in blood levels of these fatty acids are associated with a markedly elevated risk of premature birth (PTB) and early PTB.	Cetin, Irene et al., 2024 [106]
Vitamin D	Vitamin D supplementation alone has produced uncertain evidence on PE, gestational diabetes, preterm birth or nephritic syndrome. It may reduce the risk of severe postpartum hemorrhage and the risk of low birth weight.	Palacios, Cristina et al., 2024 [108]
Calcium	Calcium supplementation has been demonstrated to lower blood pressure by reducing parathyroid hormone release and intracellular calcium, which results in reduced vascular smooth muscle contractility. Consequently, it can reduce uterine smooth muscle contractility and prevent preterm labor. Prior to conception and in the early stages of pregnancy, it may have more beneficial effects on PE.	Dwarkanath, Pratibha et al., 2024 [107]

## 5. Folic Acid (FA)

Folate plays a crucial role in placental development, influencing embryogenesis and fetal growth [109,110,111], and maternal folate deficiency can adversely affect placental development, potentially affecting the risk of congenital malformations. Folate requirements during pregnancy are high due to its involvement in DNA synthesis, cell division and proliferation, placental and fetal growth and development [112]. In pregnant women, low folate levels are associated with an increased risk of neural tube defects, recurrent miscarriage, PE and preterm birth [112,113,114]. The placenta facilitates the transfer of folate to the fetus via specific transporters, including folate receptor-α (FR-α), reduced folate carrier (RFC) and proton-coupled folate transporter (PCFT), and the expression and transport capacity of these transporters can be altered by pregnancy complications and environmental factors [109,110]. The main known effect of folic acid in preventing pregnancy complications is the reduction of homocysteine levels, which could reduce oxidative stress [115]. Folate is essential for converting homocysteine to methionine and for supplying the methyl group needed to transform methionine into S-adenosylmethionine (SAM), a key methyl donor in numerous reactions such as the methylation of DNA, RNA, and proteins [112,116]. It is also involved in recycling homocysteine into methionine and in the synthesis of nucleotides like purines, pyrimidines, and thymidines, which are crucial for DNA synthesis [116,117]. When SAM levels decrease due to folate deficiency, cytotoxic homocysteine levels increase, resulting in the expression of pro-inflammatory cytokines that induce oxidative stress by promoting reactive oxygen species, leading to global DNA hypomethylation [88,112] and alterations in DNA methyltransferases (DNMTs) [118]. High maternal homocysteine levels are associated with preterm birth, low birth weight, PE, miscarriage and IUGR.

The culture of human placental trophoblast has demonstrated that FA is indispensable for several pivotal phases of placental development, including the invasion of the trophoblast, the formation of placental blood vessels, and the secretion of matrix metalloproteinases. The available evidence on the relationship between FA supplementation and placental growth across different pregnancies is limited. The primary metric utilized to assess the growth and function of the placenta is placental weight. Other morphological indicators include length, width, thickness, surface area, and so forth. An ex vivo study demonstrated that placentas cultured under low-FA conditions exhibited increased apoptosis in full-term human placental primary trophoblasts [119]. The results of animal studies have indicated that the administration of FA supplementation prior to and throughout the gestational period has been associated with an increase in several key parameters, including placental and fetal weight, maximum placental diameter, junctional and labyrinth volume, and blood vessel density [120]. This may be attributed to the fact that FA has been observed to elevate the expression levels of vascular endothelial growth factor A (VEGF-A) and placental growth factor mRNA (PIGF mRNA), which in turn has been shown to enhance blood vessel density and improve fetal placental growth restriction. Low folate concentrations have been demonstrated to increase placental vascular resistance. The placenta is susceptible to oxidative stress due to the elevated metabolic activity of placental cells. As a methyl donor in the homocysteine cycle, a deficiency of folate can result in elevated levels of homocysteine in the blood. Elevated homocysteine levels contribute to the induction of oxidative stress by promoting the production of hydrogen peroxide and superoxide free radicals. These deleterious factors impair the functionality of the placental vascular endothelium, impede the angiogenesis process of the villi, and reduce the degree of denudation of the villous capillary bed, resulting in a thinner placenta [121].

Folate supplementation during pregnancy is always recommended, often even before fertilization when planning a pregnancy.

## 6. Omega-3 Polyunsaturated Fatty Acids (PUFAs)

Maternal supplementation with omega-3 PUFAs during pregnancy increases the duration of pregnancy, thereby reducing preterm birth, improves fetal growth and reduces the risk of pregnancy complications [122,123]. Omega-3 PUFAs, which are preferentially transferred to the fetus through the placenta, play a key role in normal fetal development and placental function. Omega-3 PUFAs promote remodeling of the utero-placental architecture to facilitate increased blood flow and surface area for nutrient exchange [122] through cellular, molecular, and epigenetic pathways. These fatty acids enhance trophoblast invasion, vascular development, and nutrient transport, while also modulating gene expression through DNA methylation, histone modification, and non-coding RNAs. They can affect histone acetylation and methylation, affecting the transcriptional activity of essential genes, and can alter the expression of specific miRNAs in the placenta, which can subsequently affect placental and fetal physiology [124]. In particular, dietary docosahexaenoic acid (DHA), a vital omega-3 fatty acid, is essential for optimal placental and fetal development, especially of the brain, nervous and visual systems, and can influence global DNA methylation patterns through monocarboxylate metabolism [125,126]. DHA receives methyl groups from SAM via the action of phosphatidylethanolamine N-transferase [125]. This process is crucial for converting phosphatidylethanolamine to phosphatidylcholine, a phospholipid essential for transporting polyunsaturated fatty acids from the liver to plasma for distribution to peripheral tissues [125]. When maternal DHA levels are low, the demand for methyl groups to convert phosphatidylethanolamine to phosphatidylcholine decreases, resulting in an excess of methyl groups that enhances global methylation of placental DNA, thereby altering placental gene expression [125]. Consequently, low dietary DHA levels may increase global DNA methylation patterns. Another mechanism of omega-3 fatty acids activated during pregnancy is the reduction of inflammation and oxidative stress; an increase in placental inflammation and oxidative stress has been associated with placental disorders such as PE, IUGR and gestational diabetes mellitus [122]. Omega-3 fatty acids reduce placental inflammation by promoting the production of anti-inflammatory eicosanoids or reducing the production of pro-inflammatory eicosanoids [122,127,128] and limit placental oxidative stress by reducing the production or increasing the scavenging of reactive oxygen species (ROS) [122]. In addition, DHA has been shown to reduce oxidative damage to placental DNA and stimulate the expression of key angiogenesis factors such as VEGF [129]. Radicals, such as ROS, have the capacity to activate a methylene bridge between two adjacent C=C double bonds, a defining feature of DHA and other PUFAs. This process involves acylation of the Cδ1 site in the tryptophan of the target protein with a methylene bridge, which in turn exerts signal-regulatory activity. It remains unclear whether acylation of the methylene bridge occurs in the placenta of women with PE. However, it is hypothesized that oxidative stress caused by placental hypoxia in PE results in ROS overproduction, thereby allowing DHA and other PUFAs to alter signaling pathways crucial for placental adaptation to the maternal–fetal environment via acylation of methylene-bridged tryptophan [130]. The study conducted by Lidong Liu et al. (2024) [131] revealed a reduction in overall DHA acylation in the placenta of patients with PE. Furthermore, DHA supplementation was observed to enhance DHA acylation on protein kinase B (AKT) in HTR-8/SVneo cells. However, this effect was competitively reversed by tryptophan supplementation. These findings indicate that sufficient levels of DHA in the placenta may undergo acylation to AKT via a methylene bridge, thereby activating AKT-VEGFA signaling and promoting placental angiogenesis. Conversely, a deficiency of DHA in the placenta may impair placental angiogenesis, thereby contributing to the pathogenesis of PE [131].

## 7. Vitamin D and Calcium

Vitamin D plays a crucial role in placental development and its deficiency can lead to adverse pregnancy outcomes, including an increased risk of pre-eclampsia and gestational diabetes mellitus [91,132]. Vitamin D supplementation during pregnancy has shown potential benefits in reducing the risk of complications and improving maternal and fetal outcomes [133]. Vitamin D affects cell proliferation, differentiation, angiogenesis, fetal bone formation and immune modulation, which are critical for placental development. The optimal structure of the chorionic villi ensures proper nutrient delivery to the fetus, but in vitamin D deficiency, the chorionic villi show several structural changes, including villous edema, villous stroma fibrosis and thickening of the basement membrane of the fetal capillaries. This compromises vascular integrity and increases the distance between the fetal capillaries and the intervillous space, reducing the transfer of O_2_ from maternal to fetal blood.

These changes are associated with adverse pregnancy outcomes due to maternal vascular malperfusion [133]. In addition, vitamin D protects endothelial cells from oxidative stress and reduces the effects of exposure factors associated with pre-eclampsia. Vitamin D supplementation improves vascular elasticity and the thickness of the media and intima of blood vessels [133]. MicroRNA (miRNA) molecules are co-inducible in key placental developmental processes [134], molecules that play a role in many diseases and are involved in post-transcriptional gene expression and modulation of pathways that control organ function and differentiation [135]. It is therefore reasonable to assume that abnormal changes in the placenta of deficient women are due to a dysfunction in miRNA expression. Functional roles of miRNAs include the control of trophoblast differentiation, proliferation, invasion, migration, apoptosis, angiogenesis and cell metabolism, such as intrauterine growth restriction and pre-eclampsia [136,137]. It has been shown that vitamin D regulates cell cycle progression and has pro-differentiation and anti-proliferative effects in several cell types, including mesenchymal cells, endothelial cells, immune cells, keratinocytes, chondrocytes, osteoblasts and neural cells. Differentiation is mediated by changes in the expression of growth factors and cytokines, while proliferation effects are mediated by the induction of cell cycle inhibitors that prevent the transition from G1 to S phase of the cell cycle [138]. This suggests that the effects of vitamin D deficiency on cell differentiation and proliferation are complex. In addition, the presence of 1α-hydroxylase, an enzyme that converts 25(OH)vitaminD_3_ to the biologically active form 1,25(OH)vitaminD_3_, on placental cells indicates the importance of vitamin D in the development of a healthy pregnancy [139].

Vitamin D stimulates insulin production by preventing insulin deficiency, a central factor in the pathogenesis of gestational diabetes mellitus, and reduces pro-inflammatory cytokines present in the pre-eclamptic placenta [132]. Vitamin D exerts its biological effects by binding to and activating the vitamin D receptor (VDR), which then forms a heterodimer with retinoid X receptor (RXR) to regulate gene expression by attaching to target gene promoters [140,141]. In placental tissue of pre-eclamptic pregnancies, VDR and RXR are found to be downregulated due to DNA hypermethylation [142]. DNA methylation also influences the expression of two crucial genes for regulating vitamin D levels during pregnancy: 25(OH)-1α-hydroxylase (CYP27B1), necessary for vitamin D activation and upregulated during pregnancy, and 24-hydroxylase (CYP24A1), which inactivates vitamin D and must be downregulated during pregnancy [141,143]. The CYP24A1 promoter is hypomethylated, while CYP27B1 is unmethylated in normotensive human placenta [143], but hypermethylated in pre-eclamptic human placenta [142]. Thus, vitamin D metabolism may be influenced by epigenetic mechanisms in women with pregnancy complications, although further research is needed to explore the role of dietary vitamin D in these epigenetic modifications.

Combined with vitamin D, calcium deficiency often occurs during pregnancy. It has been shown that women with PE have impaired calcium metabolism and that supplementation can reduce the risk of developing the condition [132,143], although it may often increase the risk of preterm birth <37 weeks compared to women who have not supplemented [144]. Extensive research has explored the mechanisms of ion transfer across trophoblast cells, identifying key components such as Ca^2+^-ATPase [145], the Na^+^/Ca^2+^ exchanger (NCX) [146,147], calbindin D9K and D28K [146,148,149], L- and T-type calcium channels [150,151], and transient receptor potential vanilloid 5 and 6 (TRPV5 and TRPV6) channels [152,153]. In particular, mice deficient in NCX due to a smaller and avascular placental labyrinth layer. These specialized calcium transporters, particularly in the basal plasma membrane of syncytiotrophoblasts, ensure that the fetus remains hypercalcemic relative to the mother [154]. Detailed reviews have covered many aspects of calcium transport in the placenta. As well as being essential for fetal development, intracellular calcium in syncytiotrophoblasts regulates several functions, including hormone secretion [155,156], nitric oxide production [157] and the activity of transport proteins [158]. Studies of the role of calcium in the human placenta are carried out in vitro using placental explants, reconstituted membranes, primary trophoblast cultures from villous tissue or placenta-derived cell lines such as BeWo and JEG-3. The decidua, the maternal part of the placenta, produces prolactin and human chorionic gonadotropin in a calcium-dependent manner [155,159]. Some research suggests that calcium influx through voltage-gated calcium channels (VGCCs) is essential for hormone secretion [156,160,161], although this is controversial due to inconsistent results from patch clamp experiments. L-type calcium channel blockers such as diltiazem and verapamil modestly inhibit TRPV6-mediated calcium uptake at higher concentrations [162]. The presence of non-selective cation channels [163], store-operated [164] and receptor-operated calcium channels [165] in the human placenta has been described, but the role of calcium in specific cellular functions remains largely unknown. For example, in the liver, glycogen conversion to glucose is regulated by calcium release from intracellular stores via glycogen phosphorylase [166]. However, in mouse placental glycogen trophoblast cells, glycogen granules occupy almost all of the cytoplasm, leaving little space for organelles [167]. Therefore, it remains uncertain whether extracellular calcium influx through plasma membrane channels is necessary for glycogen conversion in trophoblast cells.

Significant changes occur in maternal vitamin D levels and calcium metabolism during pregnancy. Calcium is transferred from the mother to the fetus through the placenta. In rats, the placenta transports 25(OH)2D and 24,25(OH)2D, but not 1,25(OH)2D. Although this transplacental transport has not been extensively studied in humans, it is believed that vitamin D transfer from mother to fetus is facilitated by higher serum levels of 1,25(OH)2D in the mother compared to the fetus. During pregnancy, the synthesis of 1,25(OH)2D in the kidney increases, and both the decidua and placenta produce significant amounts of 1,25(OH)2D through the activity of the enzyme CYP27B1 [168]. Additionally, specific methylation of the placental CYP24A1 gene suppresses its transcription, leading to increased production and accumulation of 1,25(OH)2D, which reaches levels in the maternal serum that are twice as high in the third trimester compared to non-pregnant or postpartum women. The synthesis, metabolism, and functions of vitamin D during pregnancy are intricate. The human endometrial decidua produces 1,25(OH)2D and 24,25(OH)2D, while the placenta synthesizes 24,25OH2D. Notably, 24,25(OH)2D produced by the placenta accumulates in fetal bone and may play a role in fetal skeletal ossification. In the fetal lamb, 24,25(OH)2D is the main form of vitamin D, and it may support calcium absorption through the placenta and enhance skeletal ossification without increasing calcium levels in the fetal blood or urine [168]. 1,25(OH)2D and the enzyme CYP27B1 are involved in important autocrine and paracrine immune-modulatory networks during pregnancy. 1,25(OH)2D influences decidual dendritic cells and macrophages, which interact at the maternal–fetal interface to stimulate regulatory T cells. It also inhibits the release of T helper 1 (Th1) cytokines while promoting the release of T helper 2 (Th2) cytokines, which are predominant during implantation. This immune modulation likely prevents the rejection of the implanted embryo. Additionally, 1,25(OH)2D facilitates the transformation of endometrial cells into decidual cells and upregulates the expression of the homeobox A10 (*HOXA10*) gene, which is crucial for embryo implantation and early pregnancy development [169]. Once the chorioallantoic placenta is established at the end of the first trimester, the villous tissues secrete various hormones essential for maintaining pregnancy and regulating placental function. In human syncytiotrophoblasts, vitamin D receptor (VDR), CYP27B1, CYP24A1, and 1,25(OH)2D work together in an autocrine manner to regulate the expression of key placental hormones such as hCG, human placental lactogen (hPL), estradiol, and progesterone [170]. The collective data suggest that 1,25(OH)2D supports implantation, maintains a healthy pregnancy, promotes fetal growth by facilitating calcium delivery, regulates the secretion of multiple placental hormones, and limits the production of proinflammatory cytokines.

## 8. Probiotics

The use of probiotics by pregnant women in the US and Canada ranges from 1.3% to 3.6% while the likelihood of probiotic use in the Netherlands has risen to 13.7% [171]. The correct use of probiotics in women, in general, can promote a condition of eubiosis among species of the reproductive tract microbiota, which is not always in a normal homeostasis, thus improved urogenital tract health and normal physiological functions. In the field of reproductive medicine, the Lactobacillus species (*Lactobacillus reuteri* RC-14, *Lactobacillus fermentum*, *Lactobacillus gasseri*, *Lactobacillus rhamnosus*, *Lactobacillus acidophilus*, *Lactobacillus crispatus*) are of particular interest. The main representatives of probiotics that modulate fertility dysbiosis are *Lactobacillus casei* and *Lactobacillus salivarius*, *Bifidobacterium* species and *Bacillus species* [172]. The characteristics and capabilities of select probiotic strains employed to enhance reproductive function and reproductive health are elucidated in Table 5.

Studies have evaluated the effect of probiotics on maternal serum and placental morphology, although there is no direct information on the effect of probiotics on placental development. Probiotics play an important role in prenatal health, placental health and fetal development. They have been associated with a reduced risk of preterm birth and development of pre-eclampsia, improved maternal health with a role in maternal gut microbiota, lipid metabolism, vaginal infections, gestational diabetes mellitus, allergic diseases and alleviation of inflammatory conditions [173,174,175]. Table 6 presents the findings of multiple epidemiological studies conducted by diverse research teams in various countries between 2008 and 2023, with a focus on women experiencing and not experiencing complications during pregnancy [176].

It is well known that the presence of microbes in the placenta, amniotic fluid and fetal membranes influences the course of pregnancy from conception to birth, as well as the development of the infant’s immune system [203].

It is known that the gut microbiota undergoes significant changes during pregnancy from the first to the third trimester, independent of the health and diet of the pregnant woman. It changes from a diverse to a less diverse composition, with an increase in *Proteobacteria* and *Actinobacteria* [204], similar to that observed in inflammatory bowel disease and obesity [205]. These changes are associated with several adverse health effects; when women are overweight, there is a decrease in the number of *Bifidobacterium* and an increase in the number of *Bacteroides*, *Enterobacteriaceae*, *Staphylococci* and *Clostridia* [206], and infants born have higher concentrations of *Bacteroides*, *Clostridia* and *Staphylococci* and lower concentrations of *Bifidobacterium* compared to infants born to women of healthy weight [207]. In infants born to mothers with gestational diabetes, there is a significant increase in the genera *Lachnospiraceae*, *Bacteroides* and *Parabacteroides* compared to healthy non-diabetic mothers [208], with an increased risk of developing type 2 diabetes mellitus, hypertension, cardiovascular disease, renal dysfunction, pre-eclampsia, macrosomia and excess adiposity in infants later in life [209,210,211]. In addition, reduced *Bifidobacterium* and increased *Staphylococcus aureus* indicate a high risk of childhood obesity [212]. Higher numbers of *Bacteroides*, *Gardnerella*, *Mobiluncus*, *Peptostreptococcus* and *Prevotellagenerail* bacteria in the maternal placental microbiota have also been associated with preterm birth [213]. Low levels of *Bifidobacterium* and *Lactobacillus* strains in infants are associated with an increased risk of developing allergic diseases [214]. This all points to the importance of maternal biodiversity during pregnancy.

The use of probiotics with different formulations has shown positive effects on both maternal and fetal health; the use of *Lactobacillus rhamnosus* and *Bifidobacterium lactis* significantly reduced the incidence of gestational diabetes mellitus to 13% and resulted in a normal growth rate in newborns; the use of *L. acidophilus*, *L. casei* and *B. bifidum* significantly reduced fasting blood glucose, serum insulin levels, serum triglycerides, very low density lipoprotein (VLDL) and cholesterol concentrations. Pregnant women showed a reduction in insulin resistance [215]. In a review of the role of probiotics in the prevention of pre-eclampsia, the consumption of dairy products containing the probiotic strains *L. rhamnosus GG*, *L. acidophilus LA-5* and *B. lactis Bb12* was found to significantly reduce the risk of pre-eclampsia and hypertension [216]. In addition, the use of *Lactobacillus* spp., *Bifidobacterium* spp. and *Streptococcus* has been shown to induce high tolerance and a good ability to bind toxic elements such as arsenic, cadmium, mercury and lead, reducing their absorption and leading to significantly lower levels of cadmium in infant stool samples. This demonstrates the impact that probiotics can have in the fight against toxic elements, which is useful in combating various birth defects in newborns [217].

Probiotics can regulate macrophages to induce an enhanced immune response; the underlying mechanism is the enhancement of macrophage autophagy to defend against probiotic-induced infection [218]. *Lactobacillus acidophilus* and *Bacillus clausii* have been shown to be potent activators of the innate immune response in the murine macrophage cell line RAW264.7 [219]. These mediate immunostimulatory activity by interacting with both microorganism-associated molecular patterns and Toll-like receptors, which are also involved in stimulating autophagy in macrophages. However, studies on this topic are insufficient to provide a clear description of the process [220]. It is currently believed that probiotics can regulate autophagy in the placenta of normal pregnant women, with beneficial effects for the fetus and newborn, and provide a clinical basis for their use in abnormal pregnancy. The role of probiotics in mediating autophagy in humans remains poorly understood, with even less evidence available regarding the effects of probiotics on placental autophagy in pregnant women. The basal autophagy level of the placental trophoblast plays a pivotal role in the entire pregnancy process, including embryo implantation and placental vascular recasting. If autophagy is excessive, it can transform from a protective mechanism to a harmful one, hindering the metabolic functioning of placental trophoblasts and leading to certain diseases of placental origin, such as pre-eclampsia and fetal growth restriction. The expression of autophagy proteins LC3 and Beclin1 is elevated in pregnant women with pre-eclampsia in comparison to those without the condition [38]. The pathological process of pre-eclampsia is accompanied by impaired functioning of placental trophoblastic cells, insufficient remodeling of uterine spiral arteries and oxidative damage. It is conceivable that an increase in placental autophagic activity may facilitate trophoblast adaptation to pathological changes, thereby maintaining bioenergetic homeostasis and removing damaged organelles. In a study published in 2020, Ping Yang and colleagues [46] demonstrated that probiotic supplementation can induce a reduction of the autophagy-related protein Beclin1 at the mRNA level in placentas. They proposed that clinical supplementation of normal pregnant women with probiotics may prevent the onset of placenta-derived diseases. However, the assessment of specific clinical effects still requires multi-sample randomized controlled trials (RCTs) for confirmation [46].

A study has identified the role of probiotic interactions in the DNA methylation pattern of genes associated with weight gain and obesity in pregnant women and their infants. The women received probiotic capsules containing *L. rhamnosus* GG ATCC-53103 and *B. lactis* Bb12, and after blood samples were taken from the mothers and infants to assess the DNA methylation status of gene promoters, significantly reduced levels of DNA methylation were observed in 37 gene promoters in the women and 68 gene promoters in the infants. In addition, the DNA methylation pattern of five genes was similar between mothers and infants: complement component (C3), insulin-like growth factor binding protein 1 (IGFBP1), IL-5, myosin heavy chain 11 (MYH11) and solute carrier family 6 member 5 (SLC6A5). This confirms the role of probiotic interactions in modulating the DNA methylation status of genes associated with weight gain and obesity, highlighting the importance of using probiotics to address these issues [221].

## 9. Placental Microbiome

The existence or non-existence of the placental microbiome is a matter of debate [222]. The presence of bacterial deoxyribonucleic acid (DNA) in the placenta does not necessarily guarantee the presence of a thriving microbial community; rather, the presence of bacterial DNA or microbial cell wall components could induce a host response [223]. A study by Menon et al. [224] isolated 100–150 nm bacterial extracellular vesicles (BEV) from the placenta, which could be confused with the placental microbiome as they carry bacterial DNA and other components such as proteins, peptidoglycans, lipopolysaccharides, enzymes and ribonucleic acid (RNA). The authors state that these bacterial extracellular vesicles are normal components during pregnancy, derived from the microbiome that resides at different sites in the maternal body (oral cavity, skin, vagina, intestine, urogenital and respiratory tract) and reach the placenta by hematogenous route. They can cause low-grade inflammation in the placenta but do not affect the fetus or the course of pregnancy; they may also contribute to the development of the fetal immune system and protect the offspring against subsequent bacterial infections [224]. DNA-based studies then provide evidence for the presence of a low biomass endogenous microbial community within the placenta [225]. They show that the microbiota of placentas from healthy term deliveries is present and has a high abundance of *Lactobacillus* sp., *Propionibacterium* spp. and members of the *Enterobacteriaceae* family [226,227]. Fewer *Lactobacillus* spp. were found in placentas from preterm births, possibly supporting a role for this genus in positive pregnancy outcomes [228]. However, it is not entirely clear how the microorganisms enter the fetal placental compartment; hypotheses include that the microorganisms ascend from the vagina, that maternal dendritic cells sample the bacteria from the gut lumen, which are internalized and transported to the placenta, or that they enter through the blood supply to seed the placental microbiota [229]. Of the various hypotheses, the evidence for vaginal translocation is strong, in part because the *Lactobacillus*-dominant microbiota is similar to the *Lactobacilli* spp. present in the vagina, which correlates positively with gestational age [228,230]. The hypothesis of bacterial translocation by dissemination in the bloodstream is thought to be enhanced during pregnancy and lactation, when weak intercellular junctions in the intestinal and oral mucosa allow the transfer of low numbers of bacteria into the circulation [231]. But what is the role of the placental microbiota and can its composition be modified by supplementation? It is possible that the placental microbiota has a previously unrecognized role in early innate immune development as a source of antigenic determinants. The presence of bacteria does not lead to adverse pregnancy outcomes, further supporting the beneficial role of the interaction of the placental microbiome with the mother and fetus [232]. The composition of the placental microbiota can indeed be influenced by dietary supplementation. During pregnancy, any microbial supplementation or bacterial suppression may alter the maternal microbiome. Probiotics or antibiotics may alter the composition of the placental microbiota, although this is not entirely clear as direct studies are lacking. It is clear that probiotic foods support beneficial commensals in the gut microbiota, but it is unclear whether these effects extend to the placental microbiota [213].

## 10. Discussion

Micronutrient deficiencies are known to have a negative impact on maternal health and pregnancy outcomes. Deficiencies of several micronutrients, rather than just one, are linked to adverse effects on pregnancy outcomes, so addressing one deficiency is not enough if others persist. The effectiveness of nutraceuticals is still uncertain, as no studies have tested the effects of supplementation to meet all nutritional needs. Furthermore, most studies on supplementation have been conducted in industrialized countries, where deficiencies are less common, and this could lead to an underestimation of the benefits of supplementation in developing countries, where deficiencies are more common.

The potential mechanisms through which micronutrient intake during the pre-conceptional period affects the development of pregnancy must be clearly defined and understood. It can be concluded from the above that further experimental and interventional studies in humans are required in order to adjust the recommended daily values and, consequently, the recommended pre-conceptional diet for the mother. It is crucial to facilitate integration through the dissemination of health education and the implementation of health campaigns targeting women of reproductive age [233].

No documented cases of mortality or serious adverse effects have been reported in connection with the safe use of probiotics during pregnancy. The majority of adverse effects reported were of a gastrointestinal nature, including increased vaginal discharge and alterations in stool consistency. However, these were typically mild and responsive to lifestyle modifications.

A meta-analysis of 11 studies revealed the occurrence of 20 distinct adverse effects in mothers during the third trimester, including nausea, vomiting and diarrhea. It is important to note that these effects are common during pregnancy and may not be directly related to probiotic use. Furthermore, documented benefits have been observed in the prevention of gestational diabetes, mastitis, preterm delivery and infantile atopic dermatitis during and after pregnancy. Improved glucose metabolism and a reduction in inflammatory events have been observed, which suggests that the benefits may outweigh the minimal risks [234]. A significant obstacle to further research is the lack of understanding of the underlying mechanisms responsible for adverse effects. The studies reviewed frequently involved complex interventions comprising multiple probiotic strains, which makes it challenging to determine the precise causes of adverse effects.

Moreover, the dearth of knowledge among healthcare professionals regarding the potential benefits of probiotics may impede their ability to recommend them. It is thus evident that there is a necessity for the development of tools that facilitate the translation of scientific data into readily comprehensible information for both consumers and healthcare professionals. Further research is required that focuses on individual strains, dosages and the timing of administration. It is recommended that future research address these gaps in order to provide clearer guidance on the safety of probiotics during pregnancy and lactation.

Nevertheless, it can be stated that the administration of probiotics during pregnancy is currently considered to be relatively safe, with minimal risks and potential health benefits [234] (Table 7).

## 11. Conclusions

It is well known that maternal nutritional status during pregnancy can influence pregnancy outcomes through various mechanisms, including epigenetic mechanisms. The use of nutraceuticals, by correcting nutritional deficiencies, should help to prevent pregnancy disorders. Nutraceuticals, including probiotics and prebiotics, have been shown to modulate the maternal microbiota, which in turn ensures proper placental development and good health for both the pregnant woman and the unborn child.

The manipulation of the microbial communities can, therefore, represent a useful strategy in the treatment of these pathologies, considering that current pharmacological therapies are often insufficient and have several side effects. In this regard, the emphasis on microbiota composition for preventive and therapeutic purposes is constantly expanding.

For example, supplementation with specific strains of probiotics is thought to alter the composition of the maternal gut microbiota and influence the microbial environment of the placenta through translocation or immune-mediated mechanisms. Understanding and harnessing the ability to modulate the placental microbiota through dietary supplementation may open new avenues for prevention. By promoting a beneficial microbial environment within the placenta, it may be possible to enhance fetal immune development, potentially reducing the risk of infection and other immune-related disorders later in life. This approach is consistent with the broader perspective of using nutraceuticals to support the overall health of both mother and fetus and highlights the need for further research to optimize these interventions.

The choice to utilize probiotics or other nutraceuticals during pregnancy should be made on the foundation of a comprehensive understanding of the potential benefits and risks, in alignment with regulatory guidelines. A decision-making process may be undertaken in a number of ways, including the following:

One may consider three principal avenues for decision-making:Consulting with health professionals;Evaluating scientific evidence;Considering the specific needs of the individual.

## Figures and Tables

**Table 1 ijms-25-09688-t001:** General features of nutraceuticals.

	Details
Definition	Foods or parts of foods containing bioactive elements with physiological and medicinal effects that provide health benefits and prevent disease.
Examples	Prebiotics, probiotics, fiber, fatty acids, antioxidants, spices, herbs, nutrients, and supplements.
Difference from Pharmaceuticals	Considered more food-like and less drug-like.
Clinical Evidence	Health benefits supported by clinical evidence from human studies, particularly positive effects reported in meta-analyses.
Bioactive Elements	Polyphenolic compounds, isoprenoids, minerals, amino acid derivatives and fatty acids.
Health Benefits	Positive effects on cardiovascular, immune and nervous systems; role in infections, cancer, and obesity; useful in preventing acute and chronic diseases.
Current Regulations	US, UK, and Europe have streamlined regulations; India has a nascent but promising regulatory landscape; Canada allows free marketing of herbal remedies, homeopathic medicines, traditional medicines, probiotics, essential fatty acids, and amino acids.
Popular Nutraceuticals	Fish oil, prebiotics, probiotics, cranberry, garcinia, ginkgo biloba, ginseng, green tea, omega-3 fatty acids, red yeast rice and turmeric.
Demographic Factors	More women take dietary supplements than men, but male athletes take more supplements than female athletes. Older age, gender, education level, affordability and employment status influence supplement use.

**Table 2 ijms-25-09688-t002:** Some of the bacteria commonly present in probiotics.

Bacteria	Description
Lactobacillus species	
*Lactobacillus acidophilus*	Supports intestinal health; may reduce diarrhea risk.
*Lactobacillus rhamnosus*	Supports immune function; may reduce respiratory infections.
*Lactobacillus casei*	Helps maintain gut microbial balance.
*Lactobacillus plantarum*	Benefits digestive health and immune system.
*Lactobacillus paracasei*	Supports gastrointestinal health.
Bifidobacterium species	
*Bifidobacterium bifidum*	Improves intestinal health; may aid in IBS management.
*Bifidobacterium breve*	Supports gut health and immune function.
*Bifidobacterium longum*	Helps maintain gut barrier integrity; supports digestion.
Saccharomyces species	
*Saccharomyces boulardii*	Proven efficacy in treating antibiotic-associated diarrhea; supports immune function.
Others	
*Streptococcus thermophilus*	Used in yogurt fermentation; may aid lactose digestion.
*Enterococcus faecium*	Contributes to gut health and microbiome balance.
*Bacillus coagulans*	Produces lactic acid; supports digestive health.

**Table 3 ijms-25-09688-t003:** FDA regulations on probiotics.

Regulatory Category	Description	FDA Requirements
Drug or Biological Product	Intended for use as drugs to treat, cure, prevent, mitigate, or diagnose disease.	Must undergo FDA approval as biological products, similar to other drugs.
Dietary Supplements	Marketed as dietary supplements, not requiring pre-approval by the FDA. Manufacturers must notify the FDA of product claims and comply with “new dietary ingredient” regulations.	Subject to regulations ensuring safety and proper labeling post-market.
Food or Food Ingredient	Intended for use as a food or food ingredient. FDA regulates through post-market controls related to adulteration.	Focuses on ensuring products are safe and properly labeled after market placement.
Medical Food	Marketed as medical foods, specifically formulated for the dietary management of a specific medical condition.	No pre-market approval required; must meet FDA standards for medical foods.

**Table 5 ijms-25-09688-t005:** Characteristics and capabilities of selected probiotic strains in the reproductive tract.

Probiotic Species	Properties/Function
*Lactobacillus rhamnosus* (*E21* and *L3*) *Lactobacillus helveticus* (*P7*, *P12*, *S7*, *U13*) *Lactobacillus salivarius* (*N30*)	- High survival during in vitro gastrointestinal passage - Adhesion to both intestinal and vaginal epithelia - Hydrophobicity - Auto-aggregation - Co-aggregation - Reduces pH - Produces organic acids (mainly acetic acid) and hydrogen peroxide (H_2_O_2_)-Inhibits Candida species growth
*Lactobacillus strain* (*SQ0048*)	- Colonizes the vaginal microflora of healthy cows - Acts as a microbiological barrier to genital pathogen infections - Adheres to specific epithelia - Produces bacteriocins
*Lactobacillus reuteri RC14* *Lactobacillus rhamnosus GR1*	- Excellent colonizing ability - Preferred for preventing urogenital tract infections - Tolerates low pH - High adherence to uroepithelial and vaginal cells - Colonizes the vagina when administered orally - Integral to the female genital tract
*Lactobacillus rhamnosus BPL005*	- Reduces pH - Produces organic acids (mainly acetic acid) - Suppresses Propionibacterium acnes and Streptococcus agalactiae growth - No signs of cytotoxicity, vaginal irritation, or allergic contact dermatitis potential
*Lactobacillus buchneri* (*DSM 32407*)	- Does not affect the viability of epithelial cells - Does not evoke a pro-inflammatory response - Improves antioxidant status - Reduces pH - Produces organic acids such as lactic acid - Produces H_2_O_2_ and bacteriocins - Produces co-aggregation molecules that block pathogen spread
*Lactobacillus reuteri ATCC PTA 6475*	- Anti-inflammatory strain
*Lactobacillus rhamnosus CICC6141* *Lactobacillus casei BL23v*	- Adheres to the gut epithelium
*Lactobacillus rhamnosus CECT8361* *Bifidobacterium longum CECT7347*	- Antioxidant and anti-inflammatory activities
*Lactobacillus gasseri OLL2809*	- Immunostimulatory activity
*Bacillus amyloliquefaciens*	- Tolerates high temperatures - Reduces pH - Improves antioxidant status
*Bacillus subtilis* (*DSM10*) *Bacillus clausii* (*DSM 8716*) *Bacillus coagulans* (*DSM 1*) *Bacillus amyloliquefaciens* (*DSM 7*)	- Safe Bacillus species with probiotic properties
*Bifidobacterium lactis V9*	- Probiotic characteristics
*Saccharomyces cerevisiae*	- Induces pathogen co-aggregation - Has antibiotic resistance profile - Possesses anti-inflammatory properties - Suppresses Candida albicans growth

**Table 6 ijms-25-09688-t006:** Synthesis of the effects of probiotics in pregnancy: maternal, fetal and neonatal health outcomes [176].

First Author (Year)	Country of Study	Study Population	Sample Size	Intervention	Control	Route of Administration	Intervention Period	Treatment Regime	Review Outcomes Reported
Aaltonen et al., 2008 [177]	Finland	Pregnant women and their infants	256	Probiotic capsules containing *Lactobacillus rhamnosus* GG and *Bifidobacterium lactis* Bb12 at a dose of 10^10^ CFU plus dietary counselling	Placebo plus dietary counselling	Oral (capsule)	20 wk (unspecified)	Not reported	PE, PTB (<37 wk), GDM
Asgharian et al., 2020 [178]	Iran	Pregnant women with a pre- or early-pregnancy BMI ≥25, aged 18 y or older, fasting blood glucose <92 mg/dL, 20–22 weeks’ gestation	130	*Streptococcus thermophilus*, *Lactobacillus delbrueckii* subsp. *bulgaricus* 10^7^ CFU/g, *Lactobacillus acidophilus*, and *Bifidobacterium lactis* Bb12	*Streptococcus thermophilus* and *Lactobacillus delbrueckii* subsp. bulgaricus 10^7^ CFU/g	Oral (yogurt)	From 24 wk gestation until delivery	100 g yogurt/day	PE, PTB (<37 wk), GDM, NNM, stillbirth
Axling et al., 2021 [179]	Sweden	Healthy nonanemic (hemoglobin ≥ 110 g/L) pregnant women, aged 18–42 y with a singleton gestation, BMI 18–30 kg/m^2^	326	Freeze-dried *Lactiplantibacillus plantarum* 299v capsule 10^10^ CFU/g, + low level iron (4.2 mg), ascorbic acid (12 mg), and folic acid (30 µg)	Placebo	Oral (capsule)	From 10–12 weeks’ gestation until delivery	Twice daily	PTB (<37 wk), PE, PROM, maternal mortality, stillbirth, maternal sepsis, LBW (<2500 g), adverse effects of intervention, gestational age at birth
Callaway et al., 2019 [180]	Australia	Pregnant women before 16 weeks’ gestation with singleton pregnancy, BMI >25.0 kg/m^2^, >18 y, <20 weeks’ gestation	433	*Lactobacillus rhamnosus* and *Bifidobacterium animalis* lactis at a dose of >1 × 10^9^ CFU/g	Placebo	Oral (capsule)	From <20 weeks’ gestation until birth	Once daily	PE, PTB (<37 wk), PTB (<34 wk), GDM, PIH, stillbirth, SGA, LBW (<2500 g), maternal ICU, gestational age at birth
Daskalakis and Karambelas, 2017 [181]	Greece	Women with PPROM between 24 and 34 wk of gestation	115	*Lactobacillus rhamnosus* and *L. gasseri* at 1 × 10^8^ CFU (Ecovag Balance capsules) plus 3 daily doses of 1 g amoxicillin and 2 daily doses of 500 mg metronidazole intravenously for 2 d, then orally for another 8 d period.	Antibiotic treatment (amoxicillin, metronidazole)	Vaginal (capsule)	From recruitment (24–34 weeks’ gestation) for 10 d	Daily (capsule number not reported)	NNM, RDS, IVH, sepsis, NEC, gestational age at birth
Ebrahimzadeh et al., 2020 [182]	Iran	High-risk diabetic pregnant women	255	500 mg probiotic capsules containing *Lactobacillus*, *Bifidium*, and *Streptococcus*	Placebo	Oral (capsule)	From recruitment (14–16 weeks’ gestation) for 12 wk	Once daily	GDM
Facchinetti et al., 2013 [183]	Italy	Pregnant women carrying a singleton pregnancy between 10–34 weeks’ gestation, diagnosed with BV, aged 18–40 y	48	*Streptococcus thermophilius*, 3 bifidobacterium strains (*B. longum*, *B. breve*, *B. infantis*), 4 lactobacilli (Acidophilus, Plantarum, Paracasei, *Delbrueckii* subsp. *bulgaricus*, at least 112 billion bacteria per capsule	Clindamycin (100 mg)	Oral (capsule)	From recruitment for 15 d	2 tablets per d for 5 d, followed by 1 tablet a d for 10 d	BV
Farr et al., 2020 [184]	Austria	Women with singleton pregnancies, who were GBS positive	82	84 mg fructo-oligosaccharides; 0.2 billion *Lactobacilli jensenii* 100 B CFU/g. 1 billion *Lactobacilli crispatus* 100 B CFU/g, 1 billion *Lactobacilli rhamnosus* 100 B CFU/g, 0.3 billion *Lactobacilli gasseri* 100 B CFU/g	Placebo	Oral (capsule)	From recruitment (33–37 weeks’ gestation) for 2 wk	Twice daily	PTB (<37 wk), PPROM, sepsis, gestational age at birth
Gille et al., 2016 [185]	Germany	Pregnant women > 18 y	320	*Lactobacillus rhamnosus* GR-1 and *Lactobacillus reuteri* RC-14 (1 × 10^9^ CFU of each strain per capsule)	Placebo	Oral (capsule)	From first trimester for 8 wk	Once daily	PTB (<37 wk), BV, maternal adverse events
Halkjær et al., 2023 [186]	Denmark	Pregnant women with obesity (BMI ≥ 30 and <35 kg/m^2^), aged > 18 y.	50	*Streptococcus thermophilus*, *Bifidobacterium breve*, *Bifidobacterium longum*, *Bifidobacterium infantis*, and *Lactobacillus acidophilus*, *Lactobacillus plantarum, Lactobacillus paracasei*, *Lactobacillus delbrueckii* subsp. *bulgaricus* in 450 billion CFU/d	Placebo	Oral (capsule)	From 14–20 weeks’ gestation until delivery	2 capsules twice daily	PE, PTB (<37 wk), GDM, PIH, SGA, gestational age at birth
Hantoushzadeh et al., 2012 [187]	Iran	Patients with symptomatic BV in the third trimester of pregnancy	310	*Lactobacillus bulgaris*, *Streptococcus thermophilus*, probiotic Lactobacillus, and *Bifidobacterium lactis* 10^7^ colonies per milliliter	Orally administered clindamycin	Oral (yogurt)	From the third trimester for 1 wk	100 g twice daily	PTB (<37 wk), PROM, BV
Liu et al., 2020 [188]	China	Pregnant women positive for vaginal GBS	155	*Lactobacillus rhamnosus* GR-1 & *Lactobacillus reuteri* RC-14 3 × 10^10^ CFU in warm water below 30 °C	No treatment	Oral (liquid)	From 34 weeks’ gestation for 2 wk	1 pack per day	PROM
Husain et al., 2020 [189]	United Kingdom	Women between 9–14 weeks’ gestation, aged ≥ 16 y	304	2.5 billion CFUs each of *Lactobacillus rhamnosus* GR-1 and *Lactobacillus reuteri*	Placebo	Oral (capsule)	From recruitment gestation until delivery	Once daily	BV, PTB (<37 wk)
Krauss-Silva et al., 2011 [190]	Brazil	Pregnant women with asymptomatic BV or intermediate vaginal infection, after 8 and before 20 weeks’ gestation	644	*Lactobacillus rhamnosus* GR-1 and *Lactobacillus reuteri* RC-14, more than 1 million bacilli of each	Placebo	Oral (capsule)	From recruitment (8–20 weeks’ gestation) for 16 wk	Once daily	PTB (<37 wk), GDM
Lindsay et al., 2014 [191]	Ireland	Pregnant women between 24–28 weeks’ gestation, fasting blood glucose ≤ 7 mmol/L, aged 18–45 y	175	*Lacticaseibacillus rhamnosus* GG (6.5 × 10^9^ CFU per capsule)	Placebo	Oral (capsule)	From recruitment (24–28 weeks’ gestation) for 4 wk	Twice daily	GDM
Mantaring et al., 2018 [192]	The Philippines	Pregnant women at 24 to 28 weeks of gestation, planning to exclusively breastfeed for at least 2 months	233	Nutritional supplement powder with probiotics (7 × 10^8^ CFU of *Bifidobacterium lactis* and 7 × 10^8^ CFU of *Lactobacillus rhamnosus*)	Nutritional supplement powder	Oral (liquid)	From 24−28 weeks gestation to 2 months after birth (minimum)	Twice daily	PE, PTB (<37 wk), PIH, SGA, maternal adverse events
Neri et al., 1993 [193]	Israel	Pregnant women with BV, in the first trimester	84	*Lactobacillus acidophilus*	Acetic acid-soaked tampon	Vaginal (yogurt)	From first trimester	Two doses daily for 7 days, regimen repeated after 1 week	BV
OijNjideka Hemphill et al., 2023 [194]	US	Women at risk for iron deficiency anemia, pregnant with singleton, <20 weeks’ gestation, 18−45 years	20	*Lactobacillus plantarum 299v* 1 × 10^10^ CFU plus prenatal vitamins containing 27 mg iron as ferrous fumarate	Placebo plus prenatal vitamins containing 27 mg iron	Oral (capsule)	From 15−20 weeks’ gestation until admission for delivery	Once daily	GDM, gestational age at birth
Okesene-Gafa et al., 2019 [195]	New Zealand	Pregnant women with a singleton pregnancy at 12−17 weeks’ gestation, BMI ≥30	230	*Lactobacillus rhamnosus* GG and *Bifidobacterium lactis* 6.5 × 10^9^ CFU. Also received healthy nutritious foods, recipes, managing cravings, and physical activity education	Placebo	Oral (capsule)	From 12−17 weeks’ gestation until delivery	Once daily	PTB (<37 wk), GDM, PIH, composite maternal morbidity, stillbirth, SGA, maternal well-being
Pellonperä et al., 2019 [196]	Finland	Women with a self-reported prepregnancy BMI ≥25 kg/m²	439	*Lacticaseibacillus rhamnosus* HN001 and *Bifidobacterium animalis* ssp. *Lactis* 420, 10^10^ CFU per capsule	Placebo	Oral (capsule)	From <18 weeks to 6 months postpartum	Once daily	PE, PTB (<37 wk), GDM, PIH, PPH, stillbirth, SGA, maternal adverse events, gestational age at birth
Petricevic et al., 2023 [197]	Austria	Pregnant women between 10+0 − 16+0 weeks with intermediate vaginal microbiota (Nugent score 4)	129	*Lactobacillus casei rhamnosus* (*Lcr regenerans*) of >10^7^ CFU/mL	No treatment	Vaginal (tablet), sustained release of 4 days	From 10−16 weeks’ gestation	One tablet on day 1 and 1 on day 5 (8 days total)	PTB (<37 wk)
Sahhaf Ebrahimi et al., 2019 [198]	Iran	Pregnant women with GDM, in the second trimester	84	*Lactobacillus acidophilus* and *Bifidobacterium lactis* in yogurt	Normal yogurt	Oral (yogurt)	From recruitment for 8 weeks	300 mg per day for 8 weeks	Gestational age at birth
Shahriari et al., 2021 [199]	Iran	Women at high risk of GDM with gestational age <12 weeks, 18-40 years, BMI 18.5−39.9	542	*Lactobacillus acidophilus* (>7.5 × 10^9^ CFU), *Bifidobacterium longum* (>1.5 × 10^9^ CFU), *Bifidobacterium bifidum* (>6 × 10^9^ CFU)	Placebo	Oral (capsule)	From 14 weeks up to 24 weeks gestational age	Once daily	PE, GDM, gestational age at birth
Si et al., 2019 [200]	China	Pregnant women with gestational diabetes, carrying a singleton pregnancy, before 12 weeks gestation	226	Naturally fermented fresh garlic soaked in distilled water, with *L. bulgaricus* (10^8^ CFU/mL)	Naturally fermented fresh garlic	Oral (fermented garlic)	From <12 weeks gestation for 40 weeks total	5 g daily	PE, PTB (<37 wk), NNM, stillbirth, RDS, LBW (<2500 g)
Slykerman et al., 2018 [201]	New Zealand	Pregnant women with a history (or partner history) of treated asthma, eczema, or hayfever, at 35 weeks gestation	512	Either *Lactobacillus rhamnosus strain* HN001 (6 × 10^9^ CFU/day) or *Bifidobacterium animalis* ssp. *Lactis strain* HN019 (9 × 10^9^ CFU/day)	Placebo	Oral (capsule)	From enrollment to 6 months postnatally if still breastfeeding	Once daily	Long-term cognitive and developmental outcomes
Wickens et al., 2017 [202]	New Zealand	Pregnant women aged 16–43 y	423	*Lactobacillus rhamnosus* HN001, daily dose of 6 × 10^9^ CFU per capsule	Placebo	Oral (capsule)	From < 15 weeks’ gestation until delivery	Once daily	GDM

**Table 7 ijms-25-09688-t007:** A comparison of the benefits and adverse effects of probiotics.

Category	Benefits	Adverse Effects
Gut Health	Supports a balanced microbiome	Mild nausea
	Reduces risk of constipation and gastrointestinal discomfort	Vomiting
Pregnancy-Related Disorders	May prevent gestational diabetes	Diarrhea
	Reduces the risk of preterm birth	Abdominal cramping
	Helps in the treatment and prevention of mastitis	Flatulence
Glucose Metabolism	Associated with better glucose control in pregnant women	Increased vaginal discharge (minimal risk)
Inflammation	May lower inflammation, benefiting both mother and fetus	Changes in stool consistency (slight changes)
Infant Health	Linked to a reduced likelihood of infantile atopic dermatitis	Taste disturbance
Overall	Contributes to improved health status for pre-pregnant, pregnant, and postpartum individuals	No serious adverse effects or mortality observed

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
