# Peer review of "Nutraceuticals in Pregnancy: A Special Focus on Probiotics"

_ijms, 2024, doi:10.3390/ijms25179688_

Round 1

Reviewer 1 Report

Comments and Suggestions for Authors

The authors present a review article that aimed to summarize key studies on the influence of dietary supplementation on placental development, with a focus on the role of probiotics in prenatal health and fetal development.

 I would like to raise the following concerns.

1.

Epidemiological information on nutraceuticals in pregnancy should be highlighted in this review article.

A summary table could enhance clarity regarding their epidemiological indices (epidemiology study: nutraceuticals in pregnancy, with a special focus on probiotics).

2.

If the authors highlight the nutritional aspects of nutraceuticals, particularly probiotics, as described in Table 2, which lists some of the bacteria commonly present in probiotics, and in Table 3, which outlines FDA regulations on probiotics, as a promising future direction for both prenatal health and fetal development during pregnancy, an informed decision-making process is important.

3.

Although the authors present some certainty regarding the nutritional aspects of nutraceuticals, uncertainties remain. These include limited or equivocal scientific evidence, individual disagreements about potential benefits and risks, and varying attitudes from physicians, dietitians, and pregnant individuals. For both prenatal health and fetal development during pregnancy, a diagram and discussion addressing the balance between these uncertainties and certainties in the nutritional aspects of nutraceuticals, particularly probiotics, are suggested.

Author Response

Dear Reviewer #1,

We truly thank you for your insightful comments on our manuscript entitled: “Nutraceuticals in
pregnancy: a special focus on probiotics” submitted for publication as Review.
Thank you very much for taking the time to review this manuscript.
We carefully addressed all your concerns, which significantly improved the quality of our
manuscript.

More specifically:
Comments 1: Epidemiological information on nutraceuticals in pregnancy should be highlighted in
this review article. A summary table could enhance clarity regarding their epidemiological indices
(epidemiology study: nutraceuticals in pregnancy, with a special focus on probiotics).

Response 1: We are grateful to the reviewer for this observation. Two additional summary tables
have been included, one pertaining to nutraceuticals as micronutrients and the other to probiotics.
These tables present data obtained from more recent years and are accompanied by comments in the
text.

Comments 2: If the authors highlight the nutritional aspects of nutraceuticals, particularly
probiotics, as described in Table 2, which lists some of the bacteria commonly present in probiotics,
and in Table 3, which outlines FDA regulations on probiotics, as a promising future direction for
both prenatal health and fetal development during pregnancy, an informed decision-making process
is important.

Response 2: In accordance with the proposal, a potential decision-making process has been
incorporated, with the underlying premise being the current scientific understanding of the subject
matter.

Comments 3: Although the authors present some certainty regarding the nutritional aspects of
nutraceuticals, uncertainties remain. These include limited or equivocal scientific evidence,
individual disagreements about potential benefits and risks, and varying attitudes from physicians,
dietitians, and pregnant individuals. For both prenatal health and fetal development during
pregnancy, a diagram and discussion addressing the balance between these uncertainties and
certainties in the nutritional aspects of nutraceuticals, particularly probiotics, are suggested.

Response 3: In this regard, a discussion paragraph has been incorporated into the text, addressing
the discordant points highlighted for nutraceuticals, and in particular probiotics. Furthermore, a
summary table has been constructed in order to facilitate comparison between the benefits and
adverse effects.

We thank again the reviewer for her/his valuable revision of our work. In light of the above
considerations, we hope that the manuscript will now be deemed suitable for publication.

Sincerely,
Angelica Perna

Reviewer 2 Report

Comments and Suggestions for Authors

Overall this is a well-written, thorough and interesting review paper on the influence of dietary supplementation (probiotics) on placental & fetal development. I only have a couple of suggestions that should be addressed:

1) regarding folate (section 5); what are the effects on the placenta? mechanism? (lines 296-302)

2) PUFA's (section 6); how do these enhance invasion of EVT's? Please expand on mechanisms.

3) what are the effects and mechanisms of calcium/vitamin D on placental cell function/differentiation?

4) lines 420-422; this section is very vague. Please be specific.

5) line 476/477; exactly how would this work? Probiotics (which?) would be provided to mom, which would cause what to the placenta and fetus??

6) The review would greatly benefit from some illustrations highlighting/summarizing proposed mechanisms or selected nutraceuticals on placenta and fetal growth/development

Author Response

Dear Reviewer #2,
We truly thank you for your insightful comments on our manuscript entitled: “Nutraceuticals in
pregnancy: a special focus on probiotics” submitted for publication as Review.
Thank you very much for taking the time to review this manuscript.
We carefully addressed all your concerns, which significantly improved the quality of our
manuscript.

More specifically:
Comments 1: Regarding folate (section 5); what are the effects on the placenta? mechanism? (lines
296-302)

Response 1: In Section 5, which addresses the role of folic acid, as recommended by the reviewer,
we have conducted a detailed examination of this in placental development.

Comments 2: PUFA's (section 6); how do these enhance invasion of EVT's? Please expand on mechanisms.

Response 2: In accordance with the recommendation of the reviewer in section 6, the text has been
augmented with the inclusion of concepts that serve to clarify the mechanism by which
polyunsaturated fatty acids (PUFA) participate in the placentation process.

Comments 3: What are the effects and mechanisms of calcium/vitaminD on placental cell
function/differentiation?

Response 3: In accordance with the reviewer's request, section 7 has been expanded to include a
more comprehensive account of the role of vitamin D and calcium in placentation.

Comments 4: Lines 420-422; this section is very vague. Please be specific.

Response 4: In response to the reviewer's suggestion, we have provided a more detailed
explanation of the concept expressed between lines 420 and 422. Additionally, we have included an
illustrative table to enhance comprehension; we have delineated the bacterial strains that play a
pivotal role in maintaining the health of the female reproductive tract.

Comments 5: Line 476/477; exactly how would this work? Probiotics (which?) would be provided
to mom, which would cause what to the placenta and fetus??

Response 5: In Section 8, on lines 476-477, we have sought to provide a more detailed clarification
of the concept that the reviewer has indicated as unclear, while also offering additional references.

Comments 6: The review would greatly benefit from some illustrations highlighting/summarizing
proposed mechanisms or selected nutraceuticals on placenta and fetal growth/development

Response 6: In order to facilitate comprehension and enhance the readability of the review, a
number of tables have been incorporated, wherein concepts have been simplified or disparate results
have been compared.

We thank again the reviewer for her/his valuable revision of our work. In light of the above
considerations, we hope that the manuscript will now be deemed suitable for publication.
Sincerely,
Angelica Perna

Round 2

Reviewer 1 Report

Comments and Suggestions for Authors

No further comment